# Bi-Temporal Anodal Transcranial Direct Current Stimulation during Slow-Wave Sleep Boosts Slow-Wave Density but Not Memory Consolidation

**DOI:** 10.3390/brainsci11040410

**Published:** 2021-03-24

**Authors:** Simon Ruch, Kristoffer Fehér, Stephanie Homan, Yosuke Morishima, Sarah Maria Mueller, Stefanie Verena Mueller, Thomas Dierks, Matthias Grieder

**Affiliations:** 1Cognitive Neuroscience of Memory and Consciousness, Institute of Psychology, University of Bern, 3012 Bern, Switzerland; simon.ruch@psy.unibe.ch; 2Department of Neurosurgery and Neurotechnology, Institute for Neuromodulation and Neurotechnology, University of Tübingen, 72076 Tübingen, Germany; 3Translational Research Center, University Hospital of Psychiatry and Psychotherapy, University of Bern, 3000 Bern, Switzerland; kristoffer.feher@upd.unibe.ch (K.F.); stephanie.homan@bli.uzh.ch (S.H.); yosuke.morishima@upd.unibe.ch (Y.M.); sarahmariamuller@gmail.com (S.M.M.); stefanie.v.mueller@gmail.com (S.V.M.); thomas.dierks@upd.unibe.ch (T.D.); 4Department of Psychiatry, Psychotherapy and Psychosomatics, University Hospital of Psychiatry, University of Zurich, 8032 Zurich, Switzerland

**Keywords:** memory consolidation, hippocampus, sleep, slow wave, transcranial direct current stimulation, temporal lobe

## Abstract

Slow-wave sleep (SWS) has been shown to promote long-term consolidation of episodic memories in hippocampo–neocortical networks. Previous research has aimed to modulate cortical sleep slow-waves and spindles to facilitate episodic memory consolidation. Here, we instead aimed to modulate hippocampal activity during slow-wave sleep using transcranial direct current stimulation in 18 healthy humans. A pair-associate episodic memory task was used to evaluate sleep-dependent memory consolidation with face–occupation stimuli. Pre- and post-nap retrieval was assessed as a measure of memory performance. Anodal stimulation with 2 mA was applied bilaterally over the lateral temporal cortex, motivated by its particularly extensive connections to the hippocampus. The participants slept in a magnetic resonance (MR)-simulator during the recordings to test the feasibility for a future MR-study. We used a sham-controlled, double-blind, counterbalanced randomized, within-subject crossover design. We show that stimulation vs. sham significantly increased slow-wave density and the temporal coupling of fast spindles and slow-waves. While retention of episodic memories across sleep was not affected across the entire sample of participants, it was impaired in participants with below-average pre-sleep memory performance. Hence, bi-temporal anodal direct current stimulation applied during sleep enhanced sleep parameters that are typically involved in memory consolidation, but it failed to improve memory consolidation and even tended to impair consolidation in poor learners. These findings suggest that artificially enhancing memory-related sleep parameters to improve memory consolidation can actually backfire in those participants who are in most need of memory improvement.

## 1. Introduction

The memory system most vulnerable to neurodegenerative disturbances is the episodic memory system, which stores and retrieves idiosyncratic experiences including spatiotemporal context [1]. Episodic memory is indeed one of the most prominent cognitive abilities to steadily decline with healthy aging [2,3], accelerated in mild cognitive impairment and Alzheimer’s disease [4,5]. A deterioration of episodic memory is also associated with several neuropsychiatric disorders including schizophrenia, bipolar disorder, and depression [6,7,8]. There is ample evidence that sleep plays a vital role in the normal functioning of episodic memory consolidation [9]. Indeed, reduced and altered slow-wave sleep (SWS) in patients with neuropsychiatric disorders may, at least in part, explain the patients’ memory deficits [10,11,12,13]. Thus, the modulation of memory-related processes during SWS is of interest as a treatment avenue to counter the deterioration of memory functions in neuropsychiatric disorders and aging.

A prominent hypothesis about the role of sleep in memory is the active system consolidation theory, which assumes a two-stage memory system that encodes and stores episodic information. In the first stage, initial encoding takes place during wakefulness, when information is temporarily stored in the hippocampus [14]. In a second stage, memory representations are reactivated by the hippocampus during sleep and are thereby transferred to the neocortex for long-term storage [15]. The transfer from the hippocampus to the neocortex is believed to be mediated by a temporal coupling between cortical slow waves (high-amplitude rhythmic activity of ~0.5–4.5 Hz), thalamocortical fast spindles (short bursts of activity of ~12–16 Hz), and hippocampal ripples short bursts of activity of 150–250 Hz originating in the hippocampus [16]. The repeated neuronal reactivation of memory representations during SWS is thought to promote consolidation by transforming and integrating episodic information to long-term storage in the neocortex [15].

Augmenting the process of long-term storage of memories in humans has been the subject of extensive research lately [17,18]. Previous studies have attempted to modulate memory-related processes during sleep through two principal approaches. Specific hippocampus-dependent memory traces can be enhanced through targeted memory reactivation [19]. To generally improve memory consolidation during sleep, however, cortically expressed slow-waves or sleep spindles can be modulated. Animal research suggests that hippocampal–neocortical interplay can be facilitated in order to influence the process of memory reactivation by modulating cortical slow waves [20]. However, the modulation of slow waves and spindles does not directly alter hippocampal activity. Indeed, the putative non-invasive modulation of slow waves in humans through auditory stimulation or through transcranial electrical stimulation (tES) has enhanced memory in some studies, e.g., [21,22] but failed to do so in others [23,24,25]. A recent study by Kim et al. [26] suggests that the mixed findings may be due to directly opposing roles of lower frequency (slow oscillations (SOs)) and higher frequency slow waves (delta waves), which may both be modulated by the stimulation with opposing effects. Specifically, while SOs seem to support the consolidation of memory, in line with the active systems consolidation hypothesis [15], delta waves may be involved in forgetting, in line with the synaptic homeostasis hypothesis [27]. The latter hypothesis proposes that delta waves serve to downscale and renormalize synapses that had been potentiated toward saturation as a result of information encoding during wakefulness.

The present study sets out to address whether a more effective modulation of sleep-associated episodic memory consolidation can be achieved by tES aimed at directly facilitating hippocampal activity during SWS, as opposed to modulating cortical slow waves or spindles. tES applied over the scalp can induce a local shift in neuronal excitability that extends to connected networks. An influence of conventional tES on deeper brain structures can be mediated through functional connectivity of the stimulated proximal area to distal subcortical structures [28]. Concerning the former approach, functional connectivity between the hippocampus and the lateral temporal cortex is increased during SWS, compared to wakefulness [29]. We attempt to facilitate hippocampal activity related to memory consolidation by applying anodal transcranial direct-current stimulation (tDCS; fixed polarity and intensity tES) bilaterally over the anterior temporal lobes during SWS. Anodal stimulation is believed to achieve a net depolarizing effect on the resting membrane potential, which overall enhances spontaneous neuronal firing [30]. While it is unclear how anodal temporal stimulation affects hippocampal activity and hippocampus-dependent memory processes if applied during sleep, substantial evidence suggests that such a tES approach improves the formation and retrieval of hippocampus-dependent episodic memories if applied during wakefulness, e.g., [31,32,33,34].

Our primary hypothesis was twofold. Firstly, based on the current literature, we hypothesized that sleep-associated episodic memory consolidation can be improved with active tDCS compared with sham. However, a difference with respect to memory-related sleep parameters due to tDCS would not be expected. Namely, assuming that the hippocampus triggers memory reactivation and consolidation during sleep, a direct enhancement of hippocampal activity during sleep using tDCS may improve memory consolidation without altering sleep architecture. The absence of an effect on memory-related sleep parameters in the presence of a memory improvement after tDCS would therefore support the conclusion that we directly altered hippocampal activity. Furthermore, modulation of memory-related sleep parameters is also not expected from the target of our stimulation. Physiological slow waves are most commonly initiated in the frontal cortex, in the transition between the dorsolateral and orbitofrontal cortices, from where they propagate across the brain [35], and the frontal cortex constitutes a preferential site for reliably eliciting slow waves with transcranial magnetic stimulation (TMS) [36]. In contrast, the temporal and occipital cortices have a low occurrence of slow waves and would not be assumed ideal targets to influence slow waves, nor their temporal coupling with spindles, which are prevalent frontally and centro–parietally. In order to assess a tDCS-induced alteration of memory-related sleep parameters, we assessed the possible effect on slow-wave amplitude and frequency of occurrence. As an exploratory analysis, we assessed slow-wave density, spindle density, and nesting of spindles in slow waves. In particular, stronger nesting of centro–parietal fast spindles (12–16 Hz) in up states of frontal slow waves has been found to be positively correlated with overnight consolidation [37]. We further explored whether individuals’ baseline memory performance before the nap affected the influence of tDCS on memory retention. Recent studies have indicated that baseline performance can moderate the behavioral effect of interventions that aim to improve memory consolidation during sleep. This has been shown in both motor [38] and cognitive domains [39], wherein initial low performers exhibited a larger stimulation-induced gain in performance [38,39,40,41,42].

## 2. Materials and Methods

### 2.1. Participants

A total of 31 healthy participants were recruited for this within-subject, crossover study. Participants were included if they were right-handed, aged from 18 to 45 years, spoke German (or Swiss German) as their primary language, and had normal or corrected to normal vision. All female participants were in the luteal phase of their menstrual cycle, considering the potential relationship between female hormones and sleep spindles [43]. In addition, participants were instructed to sleep less than 5 h the night prior to the recording sessions to provoke sufficient sleep pressure. Exclusion criteria were current or previous neurological or psychiatric disorders, severe somatic disorders, medical or neurological illness with possible influence on the brain physiology, intake of psychotropic medication or psychoactive substances, drug abuse or addiction including alcohol within the last two years, pregnancy (a pregnancy test was conducted on every female participant immediately before starting the recording procedure), currently breastfeeding, heart or head surgeries, tattoos close to the head, neck or shoulder and permanent makeup, and claustrophobia. Some of these criteria are only important for MRI studies. Since this study is a pre-study of a planned MRI study, all criteria were adopted. Moreover, participants with a score of 7 or higher in the restless legs syndrome screening questionnaire (RLSSQ) or 10 or higher in the Epworth sleepiness scale (ESS) or 36 and higher in the fatigue severity scale (FSS) were excluded [44,45,46].

Of the total 31 measured participants, we had to discard data of 13 participants because seven did not show SWS in at least one of the two recording sessions, one had an RLSSQ score higher than 7, two had an FSS score exceeding 36, and two had less than one minute of artifact-free electroencephalogram (EEG) data of non-rapid eye movement (NREM) sleep. Furthermore, one data set was discarded due to a technical failure. Thus, 18 participants (9 women) with a mean age of 23.9 years old (SD = 3.1; 21–32) were included.

All participants provided informed consent. The study was in accordance with the declaration of Helsinki. The Bern regional ethics committee approved the study (KEK-Nr. 083/14). Lastly, we had registered the study in the German registry of clinical trials (Deutsches Register Klinischer Studien, DRKS00009298).

### 2.2. Episodic Memory Task

To assess retention of hippocampus-dependent episodic memory, we used a pair-associate episodic memory task with a baseline retrieval test before sleep and a delayed retrieval test after sleep. The stimulus material comprised 80 greyscale male faces from the Facial Recognition Technology (FERET) database [47,48] associated with an occupation (e.g., mason, Maurer in German). The within-subject design required two sets of stimuli (one for the active tDCS and one for the sham tDCS session), each containing 40 stimulus pairs. In the encoding phase, participants were asked to learn the face–occupation associations that were displayed for five seconds on a monitor and to rate the ease of imagining the particular person executing the occupation. This rating was introduced to facilitate the generation of episodic memory traces rather than mere semantic associations. Despite the fixed stimulus duration, the time to provide the rating was self-determined to improve memory formation at the individual level, instead of restricting learning speed. There were two subsequent encoding runs. Immediately following the encoding, participants were shown half of the previously encoded faces and were instructed to rate first the salary they associated with the face on a four-step Likert scale and then name the associated occupation (baseline retrieval test). The salary rating served as a more sensitive measure of memory in case of floor effects in the cued recall of the actual occupation [49]. A second delayed retrieval test was administered 30 min after waking up when participants were asked to do the same as during baseline retrieval with the other half of the stimuli. The experiment was programmed in E-Prime (version 2.0, Psychology Software Tools Inc., Pittsburgh, PA, USA).

### 2.3. Transcranial Direct-Current Stimulation

An Eldith DC-Stimulator (NeuroConn GmbH, Ilmenau, Germany) served as the electrical stimulation device. We used rectangle-shaped rubber electrodes (5 × 7 cm), two anodes (current split) that were mounted underneath the EEG cap onto the scalp over the temporal lobes, and one cathode that was fixated on the participants’ neck. For active tDCS, we administered 2 mA direct current during SWS in trains of 120 s, preceded by a 5 s fade in and concluded by a 5 s fade out. The inter-stimulation interval was 30 s, which allowed determining participants’ sleep stage. If participants were no longer in SWS following a stimulation epoch, the onset of the next stimulation epoch was delayed until participants reentered SWS. A maximum of 15 stimulations were applied, resulting in a maximal total stimulation duration of 30 min (in case a participant showed less than 30 min of SWS, fewer stimulations were applied). In contrast, for sham tDCS, direct current at 2 mA was applied only for four seconds. The remaining stimulation properties were equal to active tDCS, resulting in a maximal total stimulation duration of 1 min, which is assumed to be of negligible physiological relevance.

tDCS was exclusively engaged during SWS as monitored visually by the experimenters using online EEG inspection. This approach entails a more homogeneous brain state that is stimulated across subjects and might account better for inter- and intra-subject variability, compared to offline or sleep stage-independent tDCS applications. Moreover, putative tDCS effects on memory consolidation could be attributed to stimulation during SWS, rather than to unidentified sleep stages.

#### 2.3.1. Determining the Target Location through Functional Connectivity Analysis

The hippocampi are not on the surface of the cortex but sit deep within the left and right temporal lobes. Conventional tES may alter activity in these deeper brain structures only via stimulation of functionally connected proximal regions [28]. According to the review by Svoboda et al. [50], the functional connectome of the episodic memory system comprises the hippocampus, the dorsomedial prefrontal cortex, the inferior parietal lobe, and the lateral temporal cortex. Although functional connectivity reported in Svoboda et al. [50] was based on awake resting-state data, Andrade et al. [29] found that functional connectivity between the hippocampus and the lateral temporal cortex is even increased during SWS as compared to wakefulness. Based on these findings, we assumed that applying tDCS over temporal regions would be the most straightforward approach to alter hippocampal activity.

To determine properly the optimal target location at which tDCS could enhance the hippocampal activity, we performed a functional connectivity analysis on resting-state fMRI data from an independent sample (*N* = 100) of the Human Connectome Project [51]. The dataset was composed of resting-state fMRI echo-planar images (1200 volumes; repetition time = 720 ms; echo time = 33.1 ms; field of view = 208 × 180 mm, plane resolution = 2 × 2 mm; slice thickness = 2 mm; multiband factor = 8; phase encoding direction = R/L). Artifacts were extensively removed. Images were normalized to the Montreal Neurological Institute (MNI) space [52]. We additionally applied spatial smoothing with a 6 mm full-width, half-maximum Gaussian kernel.

Next, we generated a bilateral hippocampus mask (Figure 1A) from the probabilistic atlas included in the SPM Anatomy toolbox (v1.8; http://www.fz-juelich.de/inm/inm-1/DE/Forschung/_docs/SPMAnatomyToolbox/SPMAnatomyToolbox_node.html (accessed on 20 August 2018)), [53]. In addition, we created mask images of gray matter (GM), white matter (WM), and cerebrospinal fluid (CSF) voxels in the MNI space to regress out the global fluctuation of blood oxygen level-dependent (BOLD) signals. We constructed 15 confounding regressors as follows: (1–3) time series of GM, WM, and CSF; (4–9) six head motion parameters; and (10–15) the first-order derivative of the six head motion parameters. After extracting the BOLD time series of the hippocampus and the whole brain BOLD signals, we regressed out the 15 confounding regressors and calculated voxel-wise correlations between the hippocampal time series and the whole brain. We converted Pearson’s correlation coefficients to z-scores using Fisher transformation. These transformed values were entered into a one-sample *t*-test for a random effect analysis.

The results yielded several voxel-clusters that exceeded the voxel-wise family wise error (FWE)-corrected threshold (T(1, 99) = 5.22, *p* < 0.05). Figure 1B illustrates the brain regions at the surface of the cortex that are functionally connected with the hippocampi. In addition to the postcentral gyrus, lateral occipital cortex, and middle-to-superior frontal gyrus, the cluster ranging from anterior to posterior middle and superior temporal lobe were particularly evident (for a complete list of significant clusters, see Appendix A). Taken together, the bilateral lateral temporal lobes appeared to be the best-suited target areas for our stimulation approach. Therefore, the position for the two 5 × 7 cm tDCS electrodes was over EEG 10–20 electrode T7 for the left and T8 for the right hemisphere (Figure 2A). The tDCS return electrode was placed on the neck.

#### 2.3.2. Current Flow Simulation

After the optimal target area was determined, the tDCS electrode montage was selected and verified by means of electric field modeling. Electric field distribution was modeled by means of a bioelectromagnetic simulator of the current flow into the brain (Soterix HD-Explore, Soterix Medical Inc., New York, NY, USA). The software uses a finite element method to compute the distribution of the electric field into an adult head model. By placing the tDCS return electrode on the neck, the simulation indicated higher current flow in the temporal lobes (Figure 2B,C). In addition, the simulation also showed a higher current flow in the hippocampi, suggesting a direct influence on the medial temporal lobes (Figure 2D). Since the hippocampi are located in proximity to the lateral ventricles, a direct influence of tDCS may be mediated through cerebrospinal fluid of the ventricles that can serve as a conduit [54].

We applied anodal stimulation over the temporal lobes with the aim of facilitating cortical activity [55,56,57,58], and thereby in extension reinforcing hippocampal activation [28].

### 2.4. Polysomnography

A total of 22 sintered silver chloride ring electrodes were mounted on a cap in a 10–20 fashion and then connected to a 16-bit BrainAmp Standard amplifier (Brain Products GmbH, Gilching, Germany). Impedances were maintained below 10 kΩ, the sampling rate was 1 kHz, the input range 3.28 mV, and an online band-pass filter was applied ranging from 0.1–1000 Hz. The reference electrode was placed at Cz and the ground electrode at POz. Note that due to the lateral placing of the tDCS anodal electrodes proximate to the mastoids, it was not possible to use EEG derivations recommended by the American Academy of Sleep Medicine (AASM) Manual [59]. However, sleep staging and analysis of sleep spindles and memory-related processes with a non-standard recording setup have been conducted successfully before [60,61,62]. Moreover, the ground electrode placement is at the experimenter’s discretion [63]. During the sleep recording, an online notch filter at 50 Hz was applied to suppress environmental artifacts. In addition to the EEG montage, adhesive electrodes were placed underneath each eye for the electrooculogram, under the chin for the electromyogram, and on the shoulder and chest, respectively, for the electrocardiogram. For these recordings, a bipolar ExG BrainAmp 16 was used (Brain Products GmbH, Gilching, Germany). The amplifier’s specifications were equal to those of the BrainAmp Standard, except that the sampling rate was 5 kHz.

### 2.5. Study Procedure

Prior to the experimental recordings, all participants completed several questionnaires (ESS, FSS, Pittsburgh sleep quality index form (PSQI), RLSSQ), which were evaluated before the first recording session [64]. Moreover, participants were informed about the study procedure (Figure 3) and made aware that on one recording session they would receive active tDCS and on the other sham. The stimulation condition was allocated in a double-blind, counterbalanced randomized, within-subject crossover design.

On the evening of the first recording session, participants completed the ESS again, followed by the pregnancy test for female participants. It has been speculated that some transcranial electrical stimulation effects may be mediated by transcutaneous activation of afferent nerves [65,66]. In fact, anesthetic cream under the electrodes was shown to significantly reduce entrainment effects of tACS [67]. To reduce these confounding influences and any risk for arousal when engaging the tDCS, we applied locally anesthetizing Emla Crème 5% (AstraZeneca, London, UK) to the scalp. Moreover, Guleyupoglu et al. [68] demonstrated that abolishing skin sensation improves the blinding of the participants. Next, the participants were introduced to the self-guided episodic memory paradigm. After the learning phase and the immediate baseline retrieval test, the EEG cap was mounted to the participant’s head including the tDCS electrodes. The first recording was a 6 min 40 s awake resting EEG in a seated position, with alternating eyes closed and eyes open conditions. Next, the participant lay down on the stretcher that belonged to an MR-simulator device (MRI Simulator System, Model No. PST-100355, Psychology Software Tools, Inc., Sharpsburgh, MD, USA). This way, we tested whether participants can sleep in a noisy and space-constrained MRI-environment. To conclude the setup, a mock head coil was attached, and the EEG impedances were double-checked. Before the main sleep recording, we conducted another resting EEG measurement in supine position. After that, the participants were informed that the sleep recording was to be started and that they could relax and sleep. During the sleep recording, the simulated MR acoustic noise was engaged using the SimFxTM Software (Psychology Software Tools Inc., Pittsburgh, PA, USA). Ideally, after the participants slept for a whole sleep cycle or a maximum of two hours, they were woken and released from the MR-simulator. We chose a two-hour sleep design for two reasons—first, for the planned subsequent sleep study in an MRI, a whole-night sleep design is not feasible, and second, because previous studies have shown declarative memory consolidation benefits from daytime naps [69,70]. The EEG recording was remotely monitored in real time. Each time SWS was visually detected (six slow waves within a moving 30 s window), a two-minute block of anodal or sham tDCS was triggered. At least 30 s was needed between the tDCS blocks to determine the continued presence of SWS. Once participants were awake, they were asked whether or not the true stimulation had been applied. This was performed to assess participants’ blinding regarding electrical stimulation. Furthermore, 30 min after waking up the participant, the delayed retrieval test was administered.

### 2.6. Data Processing and Analysis

Due to the post hoc exclusion of 13 participants (see Section 2.1 Participants), counterbalancing of conditions was incomplete. In six participants, active tDCS was applied in the first session, and sham in the second session, while in 12 participants, active tDCS was applied in the second and sham in the first session. We, therefore, controlled for session effects in all our analyses.

#### 2.6.1. Sleep Staging

Sleep staging was performed offline and independently by M.G. and S.H. in accordance with the AASM Manual [59]. As recommended, the sleep EEG was low-pass filtered at 0.5 Hz, high-pass filtered at 35 Hz, and a notch at 50 Hz was applied. Next, the EEG was segmented into 30-s epochs, and each epoch was staged into either wakefulness (W), NREM 1 (N1), NREM 2 (N2), NREM 3 (N3), or rapid eye movement (REM) sleep (R). Additionally, epochs contaminated by tDCS ramp-up or ramp-down artifacts were labeled as NX. Sleep stage N3 includes slow waves and is labeled SWS in this study for enhanced readability [59].

#### 2.6.2. EEG Preprocessing

EEG channels that showed a poor signal throughout the recording were discarded. Furthermore, data segments that were contaminated by tDCS artifacts were marked and excluded. Ramp up and ramp down of the tDCS caused EEG channels to saturate for a certain period of time. These saturated data were identified and discarded from the analysis by checking for maximal and minimal amplitudes of +/−3200 µV. Next, a high-pass filter of 0.5 Hz, a low-pass filter of 35 Hz, and a notch filter at 50 Hz were applied, followed by computing the average reference. As outlined above, contralateral referencing against auricular or mastoid electrodes was not advised due to the proximity of the tDCS electrodes. Nonetheless, the analysis of sleep-specific EEG events such as spindles and slow-oscillations is possible with average reference [71]. The remaining bad channels were interpolated (9.1% [sd = 7.1%] of all channels of all analyzed EEG data). A manual visual inspection was also performed, and bad intervals were marked. Then, all clean data were segmented starting from the first stimulation and finishing at the end of the recording. From this segmented data, N2 and SWS segments were selected for analysis.

All data were preprocessed using BrainVision Analyzer 2.0. All further analyses (quantification of slow-wave density, spindle density, and spindle nesting) were performed in Matlab 2017b, using the freely available toolboxes FieldTrip (v20180820; http://www.fieldtriptoolbox.org/ (accessed on 20 August 2018)); [72], EEGLAB (v14.1.2b; https://sccn.ucsd.edu/eeglab/ (accessed on 20 August 2018)); [73], and custom-made scripts and functions.

#### 2.6.3. Quantification of Slow-Wave Density

To quantify slow-wave density, we identified discrete slow-wave events over frontal electrodes as in Ruch et al. [74] for all artifact-free EEG segments that contained N2 and SWS and that followed onset of tDCS/sham stimulation. We computed the slow-wave count per minute of valid EEG data for each participant and each session.

To identify discrete slow waves, we first pooled the preprocessed EEG signal over the frontal electrodes F3, Fz, F4, Fp1, and Fp2 to obtain a robust estimate of frontal EEG activity. We then band-pass filtered this signal between 0.5 and 2 Hz to focus on slow-oscillatory activity and selected all data segments that contained artifact-free N2 and SWS sleep occurring after the onset of stimulation. We identified all potential slow-wave events in the remaining data segments of the band-pass filtered signal. A potential slow wave was defined as the data between two consecutive positive-to-negative zero crossings of the signal. Each of these events thus contained a negative and a positive half wave. For each event, we determined the duration, the trough of the negative half wave, and the peak of the positive half wave. Events were selected as slow waves if their duration ranged between 0.9 and 2 s (reflecting a frequency range between 0.5 and 1.1 Hz) and if the amplitude of the trough and the trough-to-peak amplitude exceeded 2/3 of the trough amplitudes and the trough-to-peak amplitudes of all candidate events. To compute slow-wave density (1/min), we divided the number of selected slow waves by the total duration of EEG data that were included in the analysis. We also computed the mean duration, the mean amplitude of the down state (trough), and the mean down-to-up-state amplitude (trough-to-peak) for each participant and each session.

#### 2.6.4. Quantification of Spindle Density

To quantify spindle density, we identified discrete spindle events over centro–parietal electrodes as in Helfrich et al. [75] for all artifact-free EEG segments that contained N2 and SWS and that followed the onset of tDCS/sham stimulation. We focused on centro–parietal fast spindle activity [76] because fast spindles seem to be involved in memory consolidation, e.g., [37,77]. We computed the spindle count per minute of valid EEG data for each participant and each session.

To identify discrete spindles, we first pooled the preprocessed EEG signal over the electrodes C3, Cz, C4, P3, Pz, and P4 to obtain a robust estimate of centro–parietal EEG activity. We then band-pass filtered this signal between 12 and 16 Hz and extracted the Hilbert envelope of the filtered signal to estimate the instantaneous power in the fast-spindle frequency band. We selected all data segments that contained artifact-free NREM sleep occurring after the onset of stimulation. Discrete spindles were identified in the remaining data segments whenever spindle power exceeded the 75th percentile for a duration of 0.5 up to 3 s. For each spindle event, we determined the onset time, the offset time, the center time, and the duration. To compute spindle density (1/min), we divided the number of discrete spindle events by the total duration of EEG data that were included in the analysis. We also computed the mean spindle duration for each participant and each session.

#### 2.6.5. Quantification of Spindle Nesting

Fast spindles predominantly occur during the up state of slow waves, i.e., they are nested into up states. Enhanced spindle nesting was found to improve memory consolidation during sleep [37].

To quantify the nesting of fast spindles into up states, we first computed the proportion of centro–parietal spindles that coincided with a discrete frontal slow wave (slow-wave nesting). The resulting score could range between 0 (no spindles coincided with a slow wave) and 1 (all spindles coincided with a slow wave). A spindle was said to coincide with a slow wave if its center time occurred between the starting point and the endpoint of a slow-wave event. Next, we assessed the proportion of slow-wave-associated spindles that coincided with an up state (up-state nesting). We divided the number of spindles whose center time coincided with the positive half wave (up state) of a slow wave by the number of all spindles that coincided with a slow wave (i.e., a positive or a negative half wave). The resulting score could range between 0 (all slow-wave-associated spindles were nested in down states) and 1 (all spindles were nested in up states), in which a value of 0.5 indicates no clear nesting of spindles into either up or down states.

### 2.7. Significance Tests

To test whether tDCS vs. sham significantly altered memory performance or sleep, we performed linear mixed model analyses in R (v3.6.1), with the parameters of interest (sleep/memory) as dependent variables. We modeled random intercepts for participants to account for the repeated measurement design. Models were fit using the lmer function of the lme4-package (v1.1-21). We started with intercept-only models and introduced stepwise new factors of interest. We performed the likelihood ratio test to assess whether the model fit improved with each step.

## 3. Results

Of the 18 participants, only one participant reported nausea shortly after waking up. No other adverse effects in relation to the experimental procedure were reported. Moreover, the blinding of the participants to the tDCS condition was successful. Following the active tDCS condition recording session, 9 out of 18 participants judged the condition correctly (50.0%). In the sham tDCS condition, 9 out of 17 participants (one participant judgment missing) judged the stimulation condition correctly (52.9%). A chi-square test did not yield any relationship between stimulation condition and judgment accuracy (Χ^2^ = 0.03, *p* = 0.86). A detailed overview of participants’ sleep habits over the past four weeks prior to the first recording session (self-reports, retrospectively assessed using the PSQI) can be accessed in Appendix A.

### 3.1. Effect of tDCS on Sleep Macrostructure

The interrater reliability of sleep staging was high (Cronbach’s α = 0.86; Cohen’s kappa: κ = 0.738, SE = 0.006, *p* < 0.001). REM sleep was not present in the sleep data, which can be expected in studies conducted in noisy environments and with a focus on the first sleep cycle only [78]. Total sleep time (TST) and sleep-stage duration data can be viewed in Table 1. No statistically significant differences between sham tDCS and active tDCS were found for any sleep macrostructure parameters.

### 3.2. Effect of tDCS on Sleep Microstructure

To assess whether tDCS influenced sleep microstructure, we performed several linear mixed model analyses, in each of which we used one of the sleep parameters of interest as a dependent variable. We modeled random intercepts for participants and introduced stepwise the following fixed effects into an intercept-only model: session, stimulation (tDCS vs. sham), and the interaction term session×stimulation. If active vs. sham tDCS alone had an effect on the dependent variable, the model fit should improve only if stimulation is entered as the term. We performed likelihood ratio tests to assess whether model fit increased significantly with each step.

#### 3.2.1. tDCS Increased Slow-Wave Density and Nesting of Spindles in Slow-Wave up States

tDCS increased slow-wave density by an estimated average of 3.623 waves per minute (SE = 0.893), Χ^2^(1) = 11.679, *p* < 0.001. Slow waves were more frequent and 0.042 s shorter (SE = 0.014) after tDCS, compared to sham (Χ^2^(1) = 7.284, *p* = 0.007). Although tDCS had no significant influence on spindle density (Χ^2^(1) = 2.539, *p* = 0.111), the probability of spindles coinciding with slow waves increased by 6.224% (SE = 2.167) after stimulation (Χ^2^(1) = 6.789, *p* = 0.009). This increase is probably due to the elevated slow-wave density after stimulation. Importantly, spindles that coincided with slow waves were 6.080% (SE = 2.689) more likely to center around up states than down states after tDCS, Χ^2^(1) = 4.450, *p* = 0.034. In fact, the probability to coincide with up states exceeded chance level of 50% only after tDCS (mean probability = 57.367%; *t*-test vs. chance: t(17) = 4.522, *p* < 0.001), but not after sham (51.364%, *t*(17) = 0.553, *p* = 0.587). This suggests that tDCS improved the nesting of spindles in slow-wave up states. These results are illustrated in Figure 4.

Neither the factor session nor the interaction between session and stimulation reached significance in any of the models (all Χ^2^(1) < 2.512 all *p* > 0.112). This suggests that the reported effects are due to stimulation and not due to carryover effects of repeated testing or due to incomplete counterbalancing of stimulation conditions across sessions. Participants’ spindle densities (*r* = 0.536, *p* = 0.022) and slow-wave densities (*r* = 0.612, *p* = 0.007) were significantly correlated between the tDCS and the sham condition. This validates our approaches for the detection of discrete slow waves and spindles and further confirms the view that individuals’ slow-wave and spindle activities are relatively stable over time and thus have trait-like characteristics [79].

#### 3.2.2. Controlling for Potential Confounding Variables

Slow-wave density and spindle nesting may be influenced by many factors that are not related to tDCS stimulation, such as sleep quality, or the amount of data available for analysis. If any of these factors happened to covary with the stimulation condition (e.g., consistently less sleep in the sham vs. the tDCS condition), the observed effect of stimulation on slow-wave density and spindle nesting might be due to this confounding factor and not due to application of tDCS.

We identified several potential confounding variables and tested whether these variables varied systematically between the sham and the tDCS condition. To this aim, we performed the same linear mixed models as for the sleep parameters of interest.

None of the parameters that were used to identify discrete slow waves (down-state amplitude, down-to-up-state amplitude) and sleep spindles (spindle duration) varied systematically as a function of stimulation (all Χ^2^(1) < 0.481, all *p* > 0.492). This suggests that the criteria to extract slow wave and spindle activity were similar across conditions.

There were also no significant differences between stimulation conditions for the amount of time spent in different sleep stages, the onset time of stimulation, or the amount of valid EEG data that went into our analyses (all Χ^2^(1) < 2.762, all *p* > 0.096). See Table 1 for an overview of all relevant parameters. This suggests that overall sleep quality and data quality were similar across conditions.

### 3.3. Effect of tDCS on Memory Performance

#### 3.3.1. tDCS Had No Overall Effect on Memory Retention across Sleep

If tDCS applied during the nap following learning improves memory consolidation during sleep, we would expect to observe a less prominent drop in memory performance from the pre- to the post-nap memory in the tDCS condition, compared to the sham condition. This should manifest in a significant two-way interaction between delay (pre-vs. post-nap) and stimulation (tDCS vs. sham) on memory performance. To test for this interaction, we performed a linear mixed model analysis with the dependent variable memory performance (number of remembered occupations per test). We modeled random intercepts for participants and introduced stepwise the following factors of interest into an intercept-only model to assess their relevance: (1) session (1 vs. 2), (2) delay (pre- vs. post-nap), (3) stimulation (tDCS vs. sham), and (4) delay×stimulation (i.e., the critical interaction term). Note that we include session as a factor to control for the incomplete counterbalancing of stimulation conditions across test sessions.

Memory performance increased significantly from session 1 to session 2 by an average of 1.875 remembered items (SE = 0.727; X^2^(1) = 6.494, *p* = 0.011; Figure 5). This performance increase is probably due to the change in participants’ familiarity with the task. Performance further tended to decay from the pre- to the post-nap test by an average of about 1.278 items (SE = 0.970; X^2^(1) = 3.507, *p* = 0.061). Importantly, neither the main effect of stimulation (X^2^(1) = 0.003, *p* = 0.954) nor the interaction between delay and stimulation (X^2^(1) = 0.002, *p* = 0.968) was significant, which suggests that tDCS applied during sleep had no impact on memory consolidation.

#### 3.3.2. tDCS Selectively Impaired Memory Retention after Poor Learning

We wanted to assess whether the effect of tDCS on memory retention across sleep depends on how well participants had learned the face–occupation pairs before going to sleep. Although we controlled for interindividual differences in learning ability by using a repeated measures design in which all participants are subjected to both stimulation conditions (tDCS and sham), learning performance could still vary substantially from day to day. This is suggested by the fact that there was no significant correlation between pre-nap memory performances of the tDCS and the sham condition (*r* = 0.19, *p* = 0.469), even though pre- and post-nap performance scores were highly correlated within each condition (all *r* > 0.83, all *p* < 0.001).

We assessed whether the influence of tDCS on the relative change in memory performance across sleep (post-minus pre-divided by pre-nap performance) was moderated by pre-nap performance. We used standardized pre-nap performance within each condition to obtain a relative measure for how well participants had learned on the given day. We used relative change to obtain a measure of retention rate that can be compared between participants independently of their initial performance (a performance drop of four items is more dramatic in a participant who only remembered four items before the nap than in a participant who remembered all 20 items; relative change controls for that).

If learning performance moderates the impact of tDCS on memory retention, we would expect a significant interaction between standardized pre-nap performance and condition. To test for this interaction, we performed a linear mixed model analysis with relative change as the dependent variable. We modeled random intercepts for participants and introduced stepwise the following factors of interest into an intercept-only model: (1) session (1 vs. 2), (2) standardized pre-nap memory performance, (3) condition (tDCS vs. sham), and (4) the interaction standardized pre-nap performance−condition.

While session had no significant effect on memory retention (X^2^(1) = 0.023, *p* = 0.881), standardized pre-nap performance was significantly associated with an overall loss in performance across sleep (X^2^(1) = 6.341, *p* = 0.012). Better learning performance was associated with a higher performance loss from pre- to post-nap. Memory retention tended to be impaired overall after tDCS compared to sham (X^2^(1) = 3.178, *p* = 0.075). Importantly, the interaction term between standardized pre-nap performance and condition was significant, X^2^(1) = 13.145, *p* < 0.001), suggesting that the impact of tDCS on memory retention depends on how well participants had learned before sleep.

Visual inspection of the correlation between pre-nap performance and relative change in memory performance across sleep (Figure 5) suggested that participants with poor learning showed a substantial performance gain after sleep but only if they did not receive tDCS during sleep.

#### 3.3.3. Sleep Microstructure Changes Were Not Related to Memory

Many studies have reported links between slow-wave density, spindle density, the nesting of spindles in up states, and memory consolidation [17,18]. While absolute spindle activity and slow-wave activity might merely mirror an individual’s overall learning ability and might thus have trait-like characteristics, the change in slow wave or spindle activity relative to an individual’s baseline might measure state-dependent processes, such as memory consolidation following a learning event, e.g., [80]. We, therefore, explored whether individual differences in absolute slow-wave and spindle activity and differences in the relative change (tDCS vs. sham) were related to memory.

Neither slow-wave density, nor spindle density, nor the nesting of spindles in up states was related to overall memory performance (pre-nap performance or average between pre- and post-nap performance) or the relative change in performance from pre- to post-nap. This was true both for the sham and the tDCS condition and for the average scores computed across sham and tDCS (all |*r*| < 0.42, all *p* > 0.08). Furthermore, neither the change in slow-wave density (*r* = −0.36, *p* = 0.146), nor the changes in spindle density (*r* = −0.04, *p* = 868) or in the nesting of spindles in up states (*r* = 0.01, *p* = 0.974) from sham to tDCS were related to the change in memory performance (i.e., the difference in relative change) between conditions. Hence, we could not replicate previous findings that show how slow waves and spindles relate to memory performance and memory consolidation.

## 4. Discussion

In this study, we attempted to boost sleep-associated episodic memory consolidation during an evening nap using a bi-temporal anodal tDCS protocol that aimed at facilitating the hippocampal activity. The study is the first attempt to directly influence hippocampal activity during SWS. Hippocampus has been proposed to trigger the processes of memory reactivation and consolidation during sleep [15]. We found that tDCS increased slow-wave density and the nesting of spindles in slow-wave up states. However, while these parameters are usually related to memory consolidation during sleep, tDCS did not improve memory retention across sleep. Moreover, sleep-related consolidation of hippocampus-dependent episodic memories was found to be impaired by stimulation in participants with a below-average baseline of memory retrieval.

The here-suggested tDCS setup was aimed at increasing activity and excitability in the temporal lobes and indirectly the hippocampus. Although slow waves are thought to be generated in the PFC, we observed enhanced slow-wave density and increased nesting of spindles in slow-wave up states following bi-temporal stimulation. We speculate that this increase in slow-wave activity and spindle nesting was triggered by the elevated hippocampal activity and excitability. Recent evidence indeed suggests that the hippocampus plays a major role in sleep physiology, especially in the generation of sleep slow-waves and in orchestrating the nesting of spindles in slow-waves. In fact, patients with selective bilateral damage to the hippocampus showed reduced slow-wave density and disrupted nesting of fast spindles in slow waves [81].

It may, however, be that an unspecific increase of hippocampal activity and excitability as intended with our tDCS paradigm might not be sufficient to improve the strengthening of specific memories. Although tDCS enhanced slow-wave activity and improved spindle nesting and thus theoretically paved the way for improved reactivation and replay of newly formed memories [15], we do not know whether the hippocampi actually replayed the induced memories for face–occupation pairs. Thus far, targeted reactivation of specific memories during sleep using auditory or olfactory cues seems to be the most successful approach to improve consolidation of specific memories [82,83]. The indirect effect of enhanced slow-wave activity [84,85] or hippocampal excitability (this study) on memory retention might depend on additional factors such as participants’ expectations about the future relevance of memory [86].

It is also possible that the tDCS-induced slow waves and spindles are different from the naturally occurring, endogenous slow waves and spindles that are known to mediate memory consolidation during sleep. Slow waves and spindles vary substantially with respect to their distributions and trajectories on the scalp [87]. It may be that tDCS only induced a specific set of oscillations, e.g., over temporal regions, that do not contribute to memory consolidation. This would explain why none of the sleep parameters were related to memory retention. Note that due to the low spatial resolution of the EEG in this study, we could not perform sophisticated topographic analyses of slow-wave activity.

Individual slow waves had a shorter average duration after tDCS compared to sham, suggesting an overall increase in spectral frequency. This could be due to reduced gamma-aminobutyric acid (GABA) concentrations [85,88] following anodal tDCS stimulation [89]. While slow waves in the slow-oscillation range are relevant for memory consolidation, faster slow waves in the delta range mediate synaptic downscaling [26]. The tDCS-induced shift to higher frequencies might have benefitted synaptic downscaling over memory consolidation. Downscaling can lead to an elimination of weak memories, e.g., [90], which could explain why tDCS selectively impaired consolidation in low-performing participants.

We cannot conclude from this study whether tDCS actually increased hippocampal activity. This would require fMRI or intracranial EEG. The aim of the present study design was indeed to pave the way for further studies in the MRI environment after establishing a behavioral stimulation effect. The study was therefore carried out in an MR-simulator. While the unusual setting cannot explain the observed effects on memory and sleep microstructure of the stimulation, the setting could have interfered with overall sleep quality and memory performance. Most importantly, the wide-band rhythmic acoustic stimulation of the MR-simulator (i.e., the simulated gradient noise) could have entrained oscillatory activity in the brain that interfered with sleep’s role in memory consolidation [91]. For example, Marshall et al. [92] found that entrainment to a theta frequency suppresses slow waves, speculating that a bidirectional relation between cortical networks underlying theta and slow-wave activity caused the suppressive effect. The use of repetition time (TR)-triggers in the scanner would allow assessing the degree of entrainment or to subtract the MR-induced entrainment. The use of active noise cancellation might further help. Another possibility might have been that the stimulation modulated activity in frontal regions through temporo–frontal connections, resulting in the increase in slow-wave density [93].

This study is not without limitations. First of all, we did not explore the effect of the polarity of stimulation (anodal vs. cathodal) on sleep and memory retention. Our assumptions about the modulatory effect of anodal stimulation over the temporal cortex are solely based on previous literature [31]. Furthermore, conditions were not properly counterbalanced across experimental sessions. Active stimulation was more often applied in the second session than in the first session. The increase of slow-wave density and of the nesting of spindles in slow-wave up states might thus be a consequence of improved sleep quality due to habituation to the sleep laboratory in the second session. However, we observed no significant differences in sleep architecture between the tDCS and the sham conditions, suggesting that sleep quality was comparable across conditions. Furthermore, we controlled for potential session effects in our statistical models. Admittedly, the incomplete counterbalancing reduced our statistical power to detect the effects of tDCS on memory performance. A final limitation is that memory performance was significantly better in the second session compared to the first. This suggests that participants were not given enough time to familiarize themselves with the task and the experimental setting during the first session. This might further have affected our ability to detect the potential effects of tDCS on memory retention.

## 5. Conclusions

We conclude that bi-temporal anodal tDCS administered during slow-wave sleep may increase slow-wave density and the nesting of spindles in slow-wave up states but fails to improve memory retention across sleep. Stimulation may even impair retention of weakly encoded memories, presumably by inducing neuronal activity that benefits synaptic downscaling and thus forgetting over memory consolidation during sleep.

## Figures and Tables

**Figure 1 brainsci-11-00410-f001:**
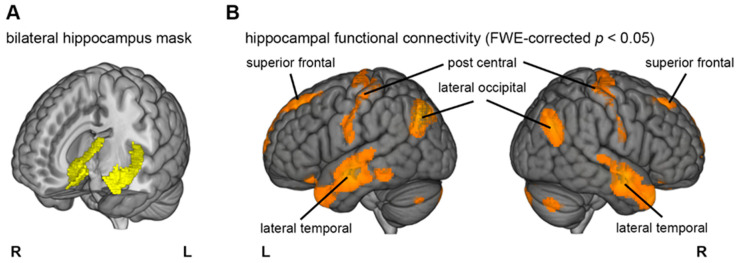
(**A**) Illustration of the hippocampal mask that was used as the seed region. (**B**) Significantly connected regions located on the surface of the cortex.

**Figure 2 brainsci-11-00410-f002:**
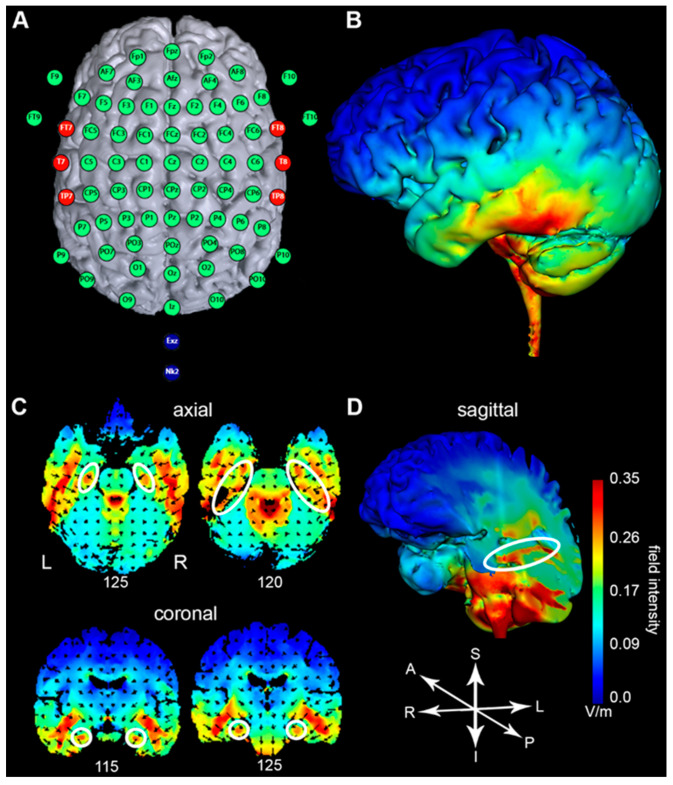
(**A**) Schematic transcranial direct-current stimulation (tDCS) montage integrated into an electroencephalogram (EEG) 10–20 montage. Note that the high-resolution EEG montage displayed in this figure was used only for anodal tDCS electrode location specification for current density simulation. This virtual EEG montage does not correspond to the recording EEG montage of this study (see Figure 3). (**B**) The lateral view of the left hemisphere shows higher electrical field intensity at the middle and inferior temporal gyri. (**C**) Axial and coronal slices illustrating higher simulated electrical field intensities in hippocampal areas (white ellipses and circles). (**D**) Sagittal cut along the hippocampus shows simulated higher electric field intensity (white ellipse). A = anterior; I = inferior; L = left; P = posterior; R = right; S = superior.

**Figure 3 brainsci-11-00410-f003:**
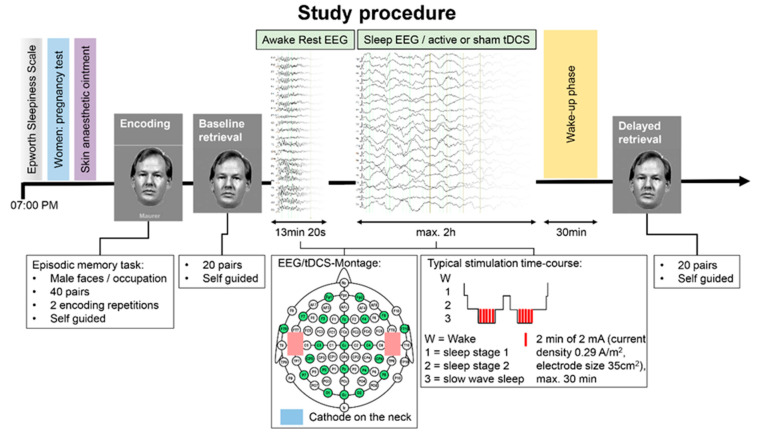
Participants underwent this study procedure twice with at least four weeks in between sessions, once with active tDCS and once with sham stimulation. Red rectangles represent anodal tDCS electrodes; the blue rectangle represents the cathodal tDCS electrode. Green shaded circles represent the 22 recording EEG channels overlaid on the 10–20 international electrode system (Fp1, Fp2, F7, F3, Fz, F4, F8, FT9, FT10, C3, Cz, C4, CP5, CP6, P7, P3, Pz, P4, P8, O1, Oz, O2).

**Figure 4 brainsci-11-00410-f004:**
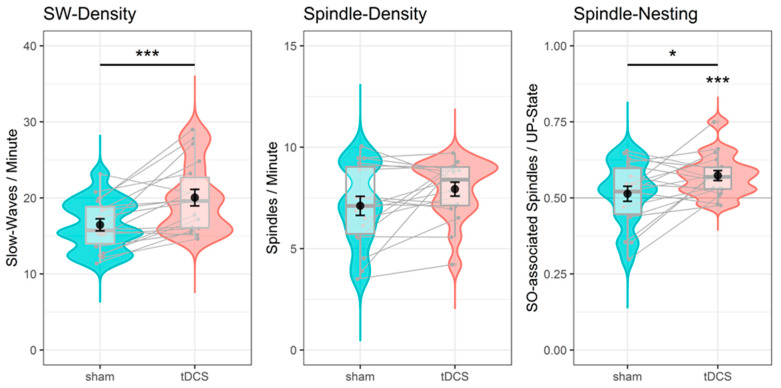
Effect of tDCS on sleep parameters. Slow-wave density (**left panel**), spindle density (**middle**), and nesting of slow-wave-associated spindles in up states (**right**). Plotted are the single participant averages (white dots), the probability densities, boxplots with median and interquartile ranges (grey), and the mean ± SE (black) for the sham (red), and the tDCS (blue) condition. tDCS significantly increased slow-wave density and the nesting of spindles in up states. Spindle nesting exceeded chance level of 0.5 only in the tDCS condition; * *p* < 0.05, *** *p* < 0.001.

**Figure 5 brainsci-11-00410-f005:**
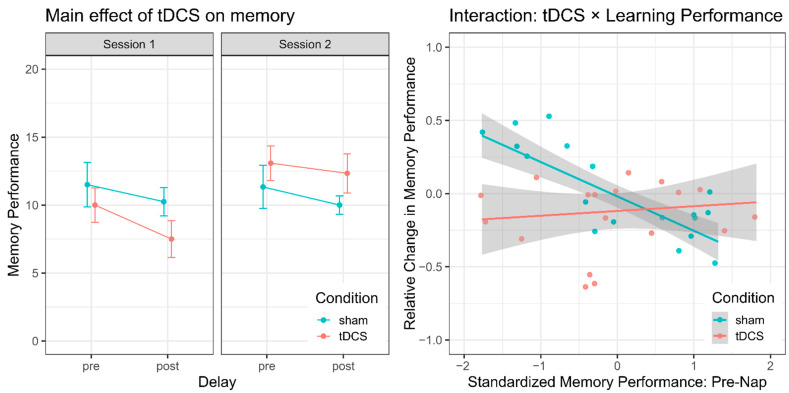
Effect of tDCS on memory performance. (**Left panel**): mean memory performance (with standard error of the mean (SEM)) for the pre- and post-nap memory test separately for the tDCS and the sham condition and for test session 1 and 2. (**Right panel**): correlation between standardized pre-nap performance and the relative change in performance (post-nap minus pre-nap divided by pre-nap performance) separately for the tDCS and the sham condition.

**Table 1 brainsci-11-00410-t001:** Descriptive statistics for all relevant parameters for the sham vs. the tDCS condition.

	Sham ^1^		tDCS		tDCS vs. Sham	
Parameter	mean	sd	mean	sd	Χ^2^(1)	*p*
Slow waves (SWs)						
SW density (SW/min)	16.466	3.333	20.046	4.628	11.687	<0.001
SW duration (s)	1.228	0.067	1.189	0.041	7.284	0.007
Down-state amplitude (μV)	−29.528	15.071	−29.372	12.543	0.003	0.958
Down-to-up state amplitude (μV)	53.966	27.422	53.605	23.602	0.008	0.931
Spindles (SPDs)						
Spindle density (SPD/min)	7.112	1.998	7.934	1.458	2.539	0.111
Spindle duration (s)	0.805	0.103	0.828	0.073	0.470	0.493
Nesting: SPDs per SWs	0.359	0.077	0.423	0.086	6.789	0.009
Nesting: SW-SPDs per Up-state	0.514	0.105	0.574	0.069	4.500	0.034
Memory performance (N remembered)						
Pre-nap	11.444	5.008	12.056	4.193	0.008	0.928
Post-nap	10.167	3.015	10.722	4.980	0.048	0.826
Change (pre-post)	−1.278	2.986	−1.333	2.679	0.119	0.730
Time per sleep stage (min)						
Time in bed	93.944	18.555	97.389	22.442	0.274	0.601
Time in Wakefulness	18.472	12.765	24.833	23.739	2.761	0.097
Time in N1	15.417	10.429	16.833	10.489	0.076	0.783
Time in N2	28.583	19.655	28.833	18.830	0.099	0.753
Time in SWS	28.167	15.360	24.917	22.538	0.189	0.664
Time in NREM (N2/SWS)	56.750	19.708	53.750	24.006	0.342	0.559
Stimulation						
Onset time	25.730	14.008	25.780	22.515	0.229	0.632
Analyzed EEG data (min)	31.550	21.884	35.644	20.834	0.558	0.445

^1^ Mean and standard deviation averaged across all 18 participants are reported separately for the sham and the tDCS condition. Significance of the effect of stimulation (tDCS vs. sham) was tested by computing the change in model fit if the factor stimulation (tDCS vs. sham) is introduced in a random intercept linear mixed model with session (1 vs. 2) as the control variable (same analyses as reported in the main text).

## Data Availability

The data presented in this study are available on request from the corresponding author. The data are not publicly available due to the local ethical regulations.

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
