# Peer review of "Bi-Temporal Anodal Transcranial Direct Current Stimulation during Slow-Wave Sleep Boosts Slow-Wave Density but Not Memory Consolidation"

_brainsci, 2021, doi:10.3390/brainsci11040410_

Round 1

Reviewer 1 Report

The manuscript is not desirable due to methodological issues. For the EEG recordings the choice of the electrodes used as reference and ground were respectively Cz and POz: if for Cz several EEG cap used such electrode as a reference, POz is not acceptable as a ground. Such montage does not allow to score correctly sleep EEG, that should be done in accordance with the rules and recommendations of the American Academy of Sleep Medicine (AASM) manual for the scoring of sleep and associated events. Such montage can bias the results relative to slow wave sleep and thus of the entire study, preventing replication and the comparison with the existing literature.

The reference and ground electrodes are the two most important electrodes. Therefore these electrodes must be placed to the most mechanically stable location. By using POz as a ground the signal is not stable and clean as in supine position participant put their weight on it. The authors were actually lucky to discard only two participants for EEG artefacts.

The abstract do not really describe correctly the study and the size of the sample is missing.

In the methods session the references for the questionnaires and scales used are missing.

Circadian factors were not taken into consideration (not even exclusion of extreme chronotypes, time of day matters!). The protocol is not individually adjusted to the participants sleep-wake schedules, thus the authors think that increasing sleep homeostasis by restricting sleep the previous night had the same effects for everyone... 

It's not clear why the authors sent sounds (i.e. the simulated MR noise) during the sleep recording. This procedure can have an impact on sleep and of course can interfere with the memory traces (and potentially can even interact with the  experimental condition, i.e. stimulation vs. sham).

Even the name of the sleep stages are not correct (the table 1 reports several sleep parameters “Time in S1”, “Time in S2”, “Time in N3”).

Reviewer 2 Report

Review for manuscript brainsci-1090048-v1 “Bi-temporal anodal transcranial direct current stimulation during slow-wave sleep boosts slow-wave density but not memory consolidation” by S. Ruch et al.

            This manuscript reports a combined tCDS and EEG study in which the authors use anodal tCDS in an attempt to modulate hippocampal activity during slow-wave sleep in humans. The authors applied anodal tCDS stimulation over bi-lateral temporal cortex under the rationale that this area has extensive connectivity with the hippocampus; thus such stimulation should propagate into the hippocampus. The goal of this stimulation was to facilitate episodic memory consolidation. The study failed to improve the latter, and indeed some evidence was observed that consolidation was impaired in slow learning participants (i.e. participants with a below average pre-sleep memory performance). However, the study did observe increased slow-wave density for stimulation versus sham conditions. Also, there was also an increased temporal coupling of spindles and slow-waves for stimulation versus sham conditions. The authors conclude that sleep-parameters are enhanced when bi-temporal anodal tCDS stimulation is applied during sleep, although such enhancement can impair memory performance of slow learners.

I found this study to be an extremely interesting application of tCDS, with a novel stimulation manipulation in the attempt to stimulate a deep structure (the hippocampus) via its connection to the superficial cortex. Moreover, the more theoretical question examined by this study (i.e. does stimulation of the hippocampus during sleep improve memory consolidation?) is interesting in its own right. Although the results were not as expected with respect to the theoretical question, the study does provide fairly compelling evidence that the sleep-parameters were indeed modulated. The case for this was bolstered by the use of a very well-controlled experimental design (sham-controlled, double-blinded, counterbalanced randomized, within-subject crossover), as well as a clear and sufficiently detailed writing style on the part of the authors. Despite the fact that it is unclear if the modulated sleep-parameters reflect a cortical or hippocampal modulation, this study does provide some foundational results upon which future research into this question can build. That said, I do have a few questions and/or clarification requests about the present manuscript that, if addressed, could make the study’s overall conclusions and impact much stronger.

Lines 173 – 193, Section 2.3. In the current version of the manuscript, the text describing the tDCS stimulation equipment and parameters is missing and instead the test repeats the previous section (Section 2.2). Please correct this and provide the correct text in this section so that the stimulation methods can be evaluated.

Line 342, Section 2.6.2. The authors state that “Remaining bad channels were interpolated.” What percentage of electrode were interpreted on average?

Lines 351 – 391, Sections 2.6.3 and 2.6.4. The authors use two different methods to quantify slow-wave and spindle density. Why did they do this when the Helfrich et al. paper they cite for the spindle density method also used a similar method for quantification of slow-waves?

Lines 625 – 626, Section 4. The authors state that “[we] cannot conclude from this study whether tDCS actually increased hippocampal activity”. The temporal lobes are known to have direct connections to the frontal lobes. Could it be possible that such direct stimulation occurred and this accounted for at least some of the present effects. Even though Figure 2 (displaying the current flow simulation results) does not show substantial stimulation-related activation in the frontal lobes, it could be the case that weak signals still had a modulatory effect. These signals might have been weaker than the signals present in temporal lobes and hippocampus, but still robust; given that the simulated temporal lobes and hippocampal current flow defines the high end of the color scale used in Figure 3, such direct temporal-frontal modulations might not be observable in such simulations.

Lines 651 – 654, Section 4. The authors state that “A final limitation is the fact that memory performance was significantly better in the second session compared to the first. This suggests that participants were not given enough time to familiarize with the task and the experimental setting during the first session”. Given this observation, could it be possible to perform a between-participants stimulation versus sham analysis just for the second session? Although as not as well controlled as the within-participants analysis presented in the manuscript, it might provide some evidence for the authors claim that this between-session performance difference “might further have affected our ability to detect potential effects of tDCS on memory retention”.

Round 2

Reviewer 1 Report

I would like to comment a couple of answers given by the authors: "...To minimize the effect of tDCS on EEG recordings, we had to put the ground and the reference electrode at maximal distance to the tDCS electrodes, which is why we selected POz and Cz respectively."  What the authors answered is not correct. As I previously commented, I can understand the choice for the reference,  but it's not true for the ground electrode: the distance to the tDCS electrodes is greater when Fpz (the typical electrode recommended by the AASM manual) is taken as ground relative to the electrode chosen  by the authors (POz). Moreover Fpz was included in the montage setting used by the authors. No sleep expert could accept such answer.  "..The head coil of the simulation scanner made contact with participants’ forehead, where the ground electrode (Fpz) is often placed. We assumed that the friction on this electrode due to potential head movements would be worse compared". I never had problems in performing acquisition in MRI using Fpz as ground. Moreover if the participant moves and the ground is placed on POz the artefact would be strong even by using foam-pads. Thus the answer is not satisfactory at all and looks like an awkward attempt to justify a wrong choice.

In the previous report, I commented the wrong terminology used for the sleep stages, that was a mix between Iber et al., 2007, and Rechtschaffen and Kales 1968. Now the authors change the terminology using again the wrong one! If the authors used the cited reference (Iber et al., 2007) to score sleep they are supposed to know how sleep stages have to be named.

The authors said that sleep scoring was done based on the AASM manuals (Iber et al., 2007; by the way, more recent versions have been released). If this is true, I would like to understand based on which derivations the scoring was done. 

At page 9, line 352 I can read "Sleep staging was performed offline and independently by MG and SW". If MG I suppose stay for Matthias Grieder, I don't understand to whom refers SW. Moreover I don't understand what does it mean that sleep staging was performed independently by the two scorers, each one scored half of the recordings?

Page 9, line 360, "To systematically exclude data contaminated by tDCS artifacts, an algorithm was applied 360 to mark such data parts." if is a validated algorithm, the name and the reference must be provided.

At page 4, line 192, is written that the tDCS was exclusively engaged during SWS. Then at page 10 the authors wrote that they identified discrete slow-wave events for segments that contained stage N2 and SWS sleep and that followed onset of tDCS/sham stimulation. Once again the methods used here is unclear and seems that there is something wrong.

Page 10, line 384 "We then band-pass filtered this signal between .5 and 2 Hz to focus on slow-oscillatory activity and selected all data segments that contained artifact-free NREM sleep occurring after onset of stimulation." I'm not sure to understand, the NREM epochs include all NREM stages...

It's not clear to me the amplitude criteria used to identify slow waves.

Relative to my comment about circadian masking factors and the answer given by the authors, I would like to highlight that in the old version of the manuscript, as well as in the new version, it was never mentioned that participants filled in a sleep diary, and an appropriate statement in the main text should be included. The legend of the table S2 reveals again the tendency of the authors to not be familiar with sleep research with the result of being not clear. Please specify in the legend that such measures refer to the weeks prior to the sleep restriction night, because observing the standard deviation I suppose that the night preceding the experiment (sleep duration 5 hr.) is not included in this calculation.

Figure 2 must be redone as it is misleading and not corresponding to the truth: there were only 22 electrodes in the present study and it should be possible to read the real electrodes labels.

Figure 3 should be improved in quality, it is not possible to read the label of the EEG channels please remove it, the same observation for the EEG/tDCS montage. It's actually only from this figure that the reader have access to the information of which electrodes were used in the montage, thus better quality is needed. Please remove the Embla creme image, it is useless.

Reviewer 2 Report

The authors have satisfactorily responded to all of my concerns. My only remaining suggestion is to report the percentage of interpolated electrodes in Section 2.6.2., as this kind of detail is needed for the reader to correctly evaluate the methods.
